# ENHANCED BAYESIAN OPTIMIZATION VIA PREFERENTIAL MODELING OF ABSTRACT PROPERTIES

## ABSTRACT

Experimental (design) optimization is a key driver in designing and discovering new products and processes. Bayesian Optimization (BO) is an effective tool for optimizing expensive and black-box experimental design processes. While Bayesian optimization is a principled data-driven approach to experimental optimization, it learns everything from scratch and could greatly benefit from the expertise of its human (domain) experts who often reason about systems at different abstraction levels using physical properties that are not necessarily directly measured (or measurable). In this paper, we propose a human-AI collaborative Bayesian framework to incorporate expert preferences about unmeasured abstract properties into the surrogate modeling to further boost the performance of BO. We provide an efficient strategy that can also handle any incorrect/misleading expert bias in preferential judgments. We discuss the convergence behavior of our proposed framework. Our experimental results involving synthetic functions and real-world datasets show the superiority of our method against the baselines.

## 1 INTRODUCTION

Experimental design is the workhorse of scientific design and discovery. Bayesian Optimization (BO) has emerged as a powerful methodology for experimental design tasks (Martinez-Cantin, 2014; Greenhill et al., 2020) due to its sample-efficiency in optimizing expensive black-box functions. In its basic form, BO starts with a set of randomly initialized designs and then sequentially suggests the next design until the target objective is reached or the optimization budget is depleted. Theoretical analyses (Srinivas et al., 2012; Chowdhury & Gopalan, 2017b) of BO methods have provided mathematical guarantees of sample efficiency in the form of sub-linear regret bounds. While BO is an efficient optimization method, it only uses data gathered during the design optimization process. However, in real-world experimental design tasks, we also have access to human experts (Swersky, 2017) who have enormous knowledge about the underlying physical phenomena. Incorporating such valuable knowledge can greatly accelerate the sample-efficiency of BO.

Previous efforts in BO literature have incorporated expert knowledge on the shape of functions (Venkatesh et al., 2019), form of trends (Li et al., 2018), model selection (Venkatesh et al., 2022) and priors over optima (Hvarfner et al., 2022), which require experts to provide very detailed knowledge about the black-box function. However, most experts understand the process in an approximate or qualitative way, and usually reason in terms of the intermediate abstract properties - the expert will compare designs, and reason as to why one design is better than another using high level abstractions. E.g, consider the design of a spacecraft shield (Whipple shield) consisting of 2 plates separated by a gap to safeguard the spacecraft against micro-meteoroid and orbital debris particle impacts. The design efficacy is measured by observing the penetration caused by hyper-velocity debris. An expert would reason why one design is better than another and accordingly come up with a new design to try out. As part of their domain knowledge, human experts often expect the first plate to shatter the space debris while the second to absorb the fragments effect. Based on these abstract intuitions, the expert will compare a pair of designs by examining the shield penetration images and ask: Does the first plate shatter better *(Shattering)*? Does the second plate absorb the fragments better *(Absorption)*? The use of such abstractions allows experts to predict the overall design objective thus resulting in an efficient experimental design process. It is important to note that measuring such abstractions is not usually feasible and only expert's qualitative inputs are available. *Incorporating such abstract properties in BO for the acceleration of experimental design process is not well explored.*

In this paper, we propose a novel human-AI collaborative approach - **B**ayesian **O**ptimization with **A**bstract **P**roperties (BOAP) - to accelerate BO by capturing expert inputs about the abstract, unmeasurable properties of the designs. Since expert inputs are usually qualitative (Nguyen et al., 2021) and often available in the form of design preferences based on abstract properties, we model each abstract property via a latent function using the qualitative pairwise rankings. We note that eliciting such pairwise preferences about designs does not add significant cognitive overhead for the expert, in contrast to asking for explicit knowledge about properties. We fit a separate rank Gaussian process (Chu & Ghahramani, 2005) to model each property. Our framework allows enormous flexibility for expert collaborations as it does not need the exact value of an abstract property, just its ranking. A schematic of our proposed BOAP framework is shown in Figure 1.

Although we anticipate that experts will provide accurate preferences on abstract properties, the expert preferential knowledge can sometimes be misleading. Therefore to avoid such undesired bias, we use two models for the black-box function. The first model uses a "main" Gaussian Process (GP) to model the black-box function in an augmented input space where the design variables are augmented with the estimated abstract properties modeled via their respective rank GPs. The second model

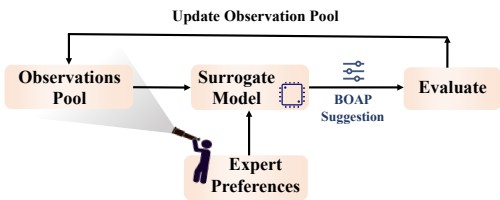

Figure 1: A schematic representation of Bayesian Optimization with Abstract Properties (BOAP)

uses another "main" GP to model the black-box function using the original design space *without* any expert inputs. At each iteration, we use predictive likelihood-based model selection to choose the "best" model that has higher probability of finding the optima.

Our contributions are: **(i)** we propose a novel human-AI collaborative BO algorithm (BOAP) for incorporating the expert pairwise preferences on abstract properties via rank GPs (Section 3), **(ii)** we provide a theoretical discussion on the convergence behavior of our proposed BOAP method (Section 3.4), **(iii)** we provide empirical results on both synthetic optimization problems and real-world design optimization problems to prove the usefulness of BOAP framework (Section 4).

## 2 BACKGROUND

### 2.1 BAYESIAN OPTIMIZATION

Bayesian Optimization (BO) (Brochu et al., 2010; Frazier, 2018) provides an elegant framework for finding the global optima of an expensive black-box function $f(\mathbf{x})$, given as $\mathbf{x}^\star \in \mathrm{argmax}_{\mathbf{x} \in \mathcal{X}} f(\mathbf{x})$, where $\mathcal{X}$ is a compact search space. BO is comprised of two main components: (i) a surrogate model (usually a Gaussian Process (Williams & Rasmussen, 2006)) of the unknown objective function $f(\mathbf{x})$, and (ii) an Acquisition Function $u(\mathbf{x})$ (Kushner, 1964) to guide the search for optima.

#### 2.1.1 GAUSSIAN PROCESS

A GP (Williams & Rasmussen, 2006) is a flexible, non-parametric distribution over functions. It is a preferred surrogate model because of its simplicity and tractability, in contrast to other surrogate models such as Student-t process (Shah et al., 2014) and Wiener process (Zhang et al., 2018). A GP is defined by a prior mean function $\mu(\mathbf{x})$ and a kernel $k : \mathcal{X} \times \mathcal{X} \rightarrow \mathbb{R}$. The function $f(\mathbf{x})$ is modeled using a GP as $f(\mathbf{x}) \sim \mathcal{GP}(0, k(\mathbf{x}, \mathbf{x}'))$. If $\mathcal{D}_{1:t} = \{\mathbf{x}_{1:t}, \mathbf{y}_{1:t}\}$ denotes a set of observations, where $y = f(\mathbf{x}) + \eta$ is the observation corrupted with noise $\eta \in \mathcal{N}(0, \sigma_\eta^2)$ then, according to the properties of GP, the observed samples $\mathcal{D}_{1:t}$ and a new observation $(\mathbf{x}_\star, f(\mathbf{x}_\star))$ are jointly Gaussian. Thus, the posterior distribution $f(\mathbf{x}_\star)$ is $\mathcal{N}(\mu(\mathbf{x}_\star), \sigma^2(\mathbf{x}_\star))$, where $\mu(\mathbf{x}_\star) = \mathbf{k}^\intercal[\mathbf{K} + \sigma_\eta^2 \mathbf{I}]^{-1}\mathbf{y}_{1:t}$, $\sigma^2(\mathbf{x}_\star) = k(\mathbf{x}_\star, \mathbf{x}_\star) - \mathbf{k}^\intercal[\mathbf{K} + \sigma_\eta^2 \mathbf{I}]^{-1}\mathbf{k}$, $\mathbf{k} = [k(\mathbf{x}_\star, \mathbf{x}_1) \cdots k(\mathbf{x}_\star, \mathbf{x}_t)]^\intercal$, and $\mathbf{K} = [k(\mathbf{x}_i, \mathbf{x}_j)]_{i,j \in \mathbb{N}_t}$.

#### 2.1.2 ACQUISITION FUNCTIONS

The acquisition function selects the next point for evaluation by balancing the exploitation vs exploration (i.e searching in high value regions vs highly uncertain regions). Some popular acquisition

functions include Expected Improvement (Mockus et al., 1978), GP-UCB (Srinivas et al., 2012) and Thompson sampling (Thompson, 1933). A standard BO algorithm is provided in Appendix § A.2.

## 2.2 RANK GP DISTRIBUTIONS

Kahneman & Tversky (2013) demonstrated that humans are better at providing qualitative comparisons than absolute magnitudes. Thus modeling latent human preferences is crucial for optimization objectives in domains such as A/B testing of web design (Siroker & Koomen, 2015), recommender systems (Brusilovski et al., 2007), players skill rating (Herbrich et al., 2006) and many more. Chu & Ghahramani (2005) proposed a non-parametric Bayesian method for learning instance or label preferences. We now discuss modeling pairwise preference relations using rank GPs.

Consider a set of $n$ distinct training instances denoted by $X = \{\mathbf{x}_i \,\forall i \in \mathbb{N}_n \mid \mathbb{N}_n = \{1, 2, \cdots, n\}\}$ based on which pairwise preference relations are observed. Let $P = \{(\mathbf{x} \succ \mathbf{x}') \mid \mathbf{x}, \mathbf{x}' \in X\}$ be a set of pairwise preference relations, where the notation $\mathbf{x} \succ \mathbf{x}'$ expresses the preference of instance $\mathbf{x}$ over $\mathbf{x}'$. E.g., the pair $\{\mathbf{x}, \mathbf{x}'\}$ can be two different spacecraft shield designs and $\mathbf{x} \succ \mathbf{x}'$ implies that spacecraft design $\mathbf{x}$ is preferred over $\mathbf{x}'$. Chu & Ghahramani (2005) assume that each training instance is associated with an unobservable latent function value $\{\bar{f}(\mathbf{x})\}$ measured from an underlying hidden preference function $\bar{f} : \mathbb{R}^d \to \mathbb{R}$, where $\mathbf{x} \succ \mathbf{x}'$, implies $\bar{f}(\mathbf{x}) > \bar{f}(\mathbf{x}')$. Employing an appropriate GP prior and likelihood, user preference can be modeled via rank GPs.

Preference learning has been used in BO literature (González et al., 2017; Mikkola et al., 2020). González et al. (2017) proposed Preferential BO (PBO) to model the unobserved objective function using a binary design preferential feedback. Benavoli et al. (2021) modified PBO to compute posteriors via skew GPs. Astudillo & Frazier (2020) proposed a preference learning based BO to model preferences in a multi-objective setup using multi-output GPs. All these works incorporate preferences about an unobserved objective function. However, in this paper, we use preference learning to model expert preferences about the intermediate abstract (auxiliary) properties. Our latent model learnt using such preferential data is then used as an input to model the main objective function.

## 3 FRAMEWORK

This paper addresses the global optimization of an expensive, black-box function, *i.e* we aim to find:

$$\mathbf{x}^\star \in \underset{\mathbf{x} \in \mathcal{X}}{\operatorname{argmax}} f(\mathbf{x}) \tag{1}$$

where $f : \mathcal{X} \to \mathbb{R}$ is a noisy and expensive objective function. For example, $f$ could be a metric signifying the strength of the spacecraft design. The motivation of this research work is to model $f$ by capturing the cognitive knowledge of experts in making preferential decisions based on the inherent non-measurable abstract properties of the possible designs. The objective here is same as that of standard BO *i.e.,* to find the optimal design $(\mathbf{x}^\star)$ that maximizes the unknown function $f$, but in the light of expert preferential knowledge on abstract properties. The central idea is to use preferential feedback to model and utilize the underlying higher-order properties that underpin preferential decisions about designs. We propose **B**ayesian **O**ptimization with **A**bstract **P**roperties (**BOAP**) for the optimization of $f$ in the light of expert preferential inputs. First, we discuss expert knowledge about abstract properties. Next, we discuss GP modeling of $f$ with preferential inputs, followed by a model-selection step that is capable of overcoming a futile expert bias in preferential knowledge. A complete algorithm for BOAP is presented in Algorithm 1 at the end of this section.

### 3.1 EXPERT PREFERENTIAL INPUTS ON ABSTRACT PROPERTIES

In numerous scenarios, domain experts reason the output of a system in terms of higher order properties $\omega_1(\mathbf{x}), \omega_2(\mathbf{x}), \ldots$ of a design $\mathbf{x} \in \mathcal{X}$. However, these abstract properties are rarely measured, only being accessible via expert preferential inputs. E.g., a material scientist designing spacecraft shield can easily provide her pairwise preferences on the properties such as shattering, shock absorption, *i.e., "this design absorbs shock better than that design"*, in contrast to specifying the exact measurements of shock absorption. These properties can be simple physical properties or abstract combinations of multiple physical properties which an expert uses to reason about the output of a system. We propose to incorporate such qualitative properties accessible to the expert in the surrogate modeling of the given objective function to further accelerate the sample-efficiency of BO.

Let $\omega_{1:m}(\mathbf{x})$ be a set of $m$ abstract properties derived from the design $\mathbf{x} \in \mathcal{X}$. For property $\omega_i$, design $\mathbf{x}$ is preferred over design $\mathbf{x}'$ if $\omega_i(\mathbf{x}) > \omega_i(\mathbf{x}')$. We denote the set of preferences provided on $\omega_i$ as $P^{\omega_i} = \{(\mathbf{x} \succ \mathbf{x}')$ if $\omega_i(\mathbf{x}) > \omega_i(\mathbf{x}') \mid \mathbf{x} \in \mathcal{X}\}$.

### 3.1.1 RANK GPs FOR ABSTRACT PROPERTIES

We capture the aforementioned expert preferential data for each of the abstract properties $\omega_{1:m}$ individually using $m$ separate rank Gaussian process distributions (Chu & Ghahramani, 2005). In conventional GPs the observation model consists of a map of input-output pairs. In contrast, the observation model of rank (preferential) Gaussian Process (GP) consists of a set of instances and a set of pairwise preferences between those instances. The central idea here is to capture the ordering over instances $X = \{\mathbf{x}_i \mid \forall i \in \mathbb{N}_n\}$ by learning latent preference functions $\{\omega_i \mid \forall i \in \mathbb{N}_m\}$. We denote such a rank GP modeling abstract property $\omega_i$ by the notation $\mathcal{GP}_{\omega_i}$.

Let $\mathcal{X} \in \mathbb{R}^d$ be a $d-$dimensional compact search space and $X = \{\mathbf{x}_i \mid \forall i \in \mathbb{N}_n\}$ be a set of $n$ training instances. Let $\boldsymbol{\omega} = \{\omega(\mathbf{x})\}$ be the unobservable latent preference function values associated with each of the instances $\mathbf{x} \in X$. Let $P$ be the set of $p$ pairwise preferences between instances in $X$, defined as $P = \{(\mathbf{x} \succ \mathbf{x}')_j$ if $\omega(\mathbf{x}) > \omega(\mathbf{x}') \mid \mathbf{x} \in X, \forall j \in \mathbb{N}_p\}$. The observation model for the rank GP distribution $\mathcal{GP}_\omega$ modeling the latent preference function $\omega$ is given as $\bar{\mathcal{D}} = \{\mathbf{x}_{1:n}, P = \{(\mathbf{x} \succ \mathbf{x}')_j \, \forall \mathbf{x}, \mathbf{x}' \in X, j \in \mathbb{N}_p\}\}$.

We follow the probabilistic kernel approach for preference learning (Chu & Ghahramani, 2005) to formulate the likelihood function and Bayesian probabilities. Imposing non-parametric GP priors on the latent function values $\boldsymbol{\omega}$, we arrive at the prior probability of $\boldsymbol{\omega}$ given by:

$$\mathcal{P}(\boldsymbol{\omega}) = (2\pi)^{-\frac{n}{2}} |\mathbf{K}|^{-\frac{1}{2}} \exp\left(-\tfrac{1}{2}\boldsymbol{\omega}^\intercal \mathbf{K}^{-1}\boldsymbol{\omega}\right) \tag{2}$$

With suitable noise assumptions $\mathcal{N}(0, \tilde{\sigma}_\eta^2)$ on inputs and the preference relations $(\mathbf{x}, \mathbf{x}')_{1:m}$ in $P$, the Gaussian likelihood function based on Thurstone (2017) is:

$$\mathcal{P}((\mathbf{x} \succ \mathbf{x}')_i | \omega(\mathbf{x}), \omega(\mathbf{x}')) = \Phi\big(z_i(\mathbf{x}, \mathbf{x}')\big) \tag{3}$$

where $\Phi$ is the c.d.f of standard normal distribution $\mathcal{N}(0, 1)$ and $z(\mathbf{x}, \mathbf{x}') = \frac{\omega(\mathbf{x}) - \omega(\mathbf{x}')}{\sqrt{2\tilde{\sigma}_\eta^2}}$. Based on Bayes theorem, the posterior distribution of the latent function given the data is given by:

$$\mathcal{P}(\boldsymbol{\omega}|\bar{\mathcal{D}}) = \frac{\mathcal{P}(\boldsymbol{\omega})}{\mathcal{P}(\bar{\mathcal{D}})}\mathcal{P}(\bar{\mathcal{D}}|\boldsymbol{\omega})$$

where $\mathcal{P}(\boldsymbol{\omega})$ is the prior distribution (Eq. (2)), $\mathcal{P}(\bar{\mathcal{D}}|\boldsymbol{\omega})$ is the probability of observing the pairwise preferences given the latent function values $\boldsymbol{\omega}$, which can be computed as a product of the likelihood (Eq. (3)) i.e., $\mathcal{P}(\bar{\mathcal{D}}|\boldsymbol{\omega}) = \prod_p \mathcal{P}((\mathbf{x} \succ \mathbf{x}')_p|\omega(\mathbf{x}), \omega(\mathbf{x}'))$ and $\mathcal{P}(\bar{\mathcal{D}}) = \int \mathcal{P}(\bar{\mathcal{D}}|\boldsymbol{\omega})\mathcal{P}(\boldsymbol{\omega})\,d\boldsymbol{\omega}$ is the evidence of model parameters including kernel hyperparameters. We find the posterior distribution using Laplace approximation and the Maximum A Posteriori estimate (MAP) $\boldsymbol{\omega}_{\text{MAP}}$ as the mode of posterior distribution. We can find the MAP using Newton-Raphson descent given by:

$$\boldsymbol{\omega}^{\text{new}} = \boldsymbol{\omega}^{\text{old}} - \mathbf{H}^{-1}\mathbf{g}|_{\boldsymbol{\omega}=\boldsymbol{\omega}^{\text{old}}} \tag{4}$$

where the Hessian $\mathbf{H} = [\mathbf{K} + \tilde{\sigma}_\eta^2\mathbf{I}]^{-1} + \mathbf{C}$, and the gradient $\mathbf{g} = \nabla_{\boldsymbol{\omega}} \log \mathcal{P}(\boldsymbol{\omega}|\bar{\mathcal{D}}) = -[\mathbf{K} + \tilde{\sigma}_\eta^2\mathbf{I}]^{-1}\boldsymbol{\omega} + \mathbf{b}$, given $b_j = \frac{\partial}{\partial \omega(\mathbf{x}_j)} \sum_p \ln \Phi(z_p)$ and $C_{ij} = \frac{\partial}{\partial \omega(\mathbf{x}_i)\partial \omega(\mathbf{x}_j)} \sum_p \ln \Phi(z_p)$.

### 3.1.2 HYPERPARAMETER OPTIMIZATION

Kernel hyperparameter ($\theta^\star$) selection is crucial to optimize the generalization performance of the GP. We perform the model selection for our rank-GPs by maximizing the corresponding log-likelihood in the light of latent values $\boldsymbol{\omega}_{\text{MAP}}$. In contrast to the evidence maximization mentioned in Chu & Ghahramani (2005) i.e., $\theta^\star = \text{argmax}_\theta \mathcal{P}(\bar{\mathcal{D}}|\theta)$, we find the optimal kernel hyperparameters by maximizing the log-likelihood ($\bar{\mathcal{L}}$) of rank GPs i.e., $\theta^\star = \text{argmax}_\theta \bar{\mathcal{L}}$. The closed-form of log-likelihood of the rank GP is given as:

$$\bar{\mathcal{L}} = -\frac{1}{2}\boldsymbol{\omega}_{\text{MAP}}^\intercal [\mathbf{K} + \tilde{\sigma}_\eta^2\mathbf{I}]^{-1}\boldsymbol{\omega}_{\text{MAP}} - \frac{1}{2}\log|\mathbf{K} + \tilde{\sigma}_\eta^2\mathbf{I}| - \frac{n}{2}\log(2\pi) \tag{5}$$

## 3.2 Augmented GP with Abstract Property Preferences

To account for property preferences in modeling $f$, we augment the input $\mathbf{x}$ of a conventional GP modeling $f$ with the mean predictions obtained from $m$ rank GPs ($\mathcal{GP}_{\omega_{1:m}}$) as auxiliary inputs capturing the property preferences $\omega_{1:m}$, *i.e.,* instead of modeling GP directly on $\mathbf{x}$ we model on $\tilde{\mathbf{x}} = [\mathbf{x}, \mu_{\omega_1}(\mathbf{x}), \cdots, \mu_{\omega_m}(\mathbf{x})]$, where $\mu_{\omega_i}$ is the predictive mean computed using:

$$\mu_{\omega_i}(\mathbf{x}) = \mathbf{k}^{\mathsf{T}}[\mathbf{K} + \tilde{\sigma}_\eta^2 \mathbf{I}]^{-1} \boldsymbol{\omega}_{\text{MAP}}$$

where $\mathbf{k} = [k(\mathbf{x}, \mathbf{x}_1), \cdots, k(\mathbf{x}, \mathbf{x}_n)]^{\mathsf{T}}$, $\mathbf{K} = [k(\mathbf{x}_i, \mathbf{x}_j)]_{i,j \in \mathbb{N}_n}$ and $\mathbf{x}_i \in X$. To handle different scaling levels in rank GPs, we normalize its output in the interval $[0, 1]$, such that $\mu_{\omega_i}(\mathbf{x}) \in [0, 1]$.

The objective function is modeled on the concatenated inputs $\tilde{\mathbf{x}} \in \mathbb{R}^{d+m}$ and we denote this function with augmented inputs $\tilde{\mathbf{x}}$ as human-inspired objective function $h(\tilde{\mathbf{x}})$. The GP ($\mathcal{GP}_h$) constructed in the light of expert preferential data is then used in BO to find the global optima of $h(\tilde{\mathbf{x}})$, given as:

$$\mathbf{x}^{\star} \in \underset{\mathbf{x} \in \mathcal{X}}{\arg\max}\, h(\tilde{\mathbf{x}}) \tag{6}$$

The observation model is $\mathcal{D} = \{(\mathbf{x}, y = h(\tilde{\mathbf{x}}) \approx f(\mathbf{x}))\}$ *i.e.,* the human-inspired objective function $h(\tilde{\mathbf{x}})$ is a simpler form compared to $f(\mathbf{x})$ due to auxiliary features in the input, and we observe the $h(\tilde{\mathbf{x}})$ via $f(\mathbf{x})$. We update $\mathcal{GP}_h$ whenever the rank GPs ($\mathcal{GP}_{\omega_i}$) are updated with new expert data.

**Incorporating Rank GP Uncertainties:** The inputs that are significantly different from the ones for which we have human preferential feedback, rank GP predictions may be inaccurate, which can adversely affect the overall optimization performance. To avoid this, we incorporate the rank GP uncertainties (which are high for such inputs). To do this, we utilize an Automatic Relevance Determination (ARD) kernel where we set the lengthscales of the augmented input dimensions in proportion to the rank GP uncertainties. Since the rank GP uncertainties are dependent on input location $\mathbf{x}$, we use a non-stationary spatially-varying kernel ($k_h$) (Venkatesh et al., 2019) that allows the lengthscale to be a parametric function of the input $\mathbf{x}$. In particular, the lengthscale function for each of the augmented input dimension (corresponding to property $\omega_i$) is set to be $l_{\omega_i}(\mathbf{x}) = \alpha \sigma_{\omega_i}(\mathbf{x})$, where $\alpha$ is the scale parameter and $\sigma_{\omega_i}(\mathbf{x})$ is the normalized standard deviation predicted using the rank GP ($\mathcal{GP}_{\omega_i}$). We set the lengthscales of the original input dimensions to a constant value $l$. We use these lengthscales in the following spatially-varying kernel function:

$$k_h(\tilde{\mathbf{x}}, \tilde{\mathbf{x}}') = \prod_{i=1}^{i=d+m} \sqrt{\frac{2l(\tilde{x}_i)l(\tilde{x}_i')}{l^2(\tilde{x}_i) + l^2(\tilde{x}_i')}} \exp\left(-\sum_{i=1}^{i=d+m} \frac{(\tilde{x}_i - \tilde{x}_i')^2}{l^2(\tilde{x}_i) + l^2(\tilde{x}_i')}\right) \tag{7}$$

where $l(\cdot)$ is the lengthscale function. This choice of lengthscale ensures that data points $\tilde{\mathbf{x}}$ with high model uncertainty have higher lengthscale on the augmented dimensions and thus are treated as less important.[1] The kernel hyperparameter set $\theta_h$ includes the constant lengthscales from the original dimensions and the $\boldsymbol{\alpha}$ parameter from the augmented input dimensions i.e., $\theta_h = [l_{1:D}, \boldsymbol{\alpha}]$. We maximize the GP log-marginal likelihood to find the optimal hyperparameter set $\theta_h^{\star}$.

## 3.3 The BOAP Algorithm

**Overcoming Inaccurate Expert Inputs:** Up to this point we have assumed that expert input is accurate and thus likely to accelerate BO. However, in some cases this feedback may be inaccurate, and potentially slow down optimization. To overcome such issues and encourage exploration we maintain 2 models, one of which is augmented by expert abstract properties (we refer to this as Human arm i.e. Arm-$\mathfrak{h}$) and an un-augmented model (we refer to this as Control arm i.e. Arm-$\mathfrak{f}$), and use maximum predictive likelihood to select the arm at each iteration.

The Control arm models $f$ directly by observing the function values at suggested candidate points. Here, we fit a standard GP ($\mathcal{GP}_f$) based on the data collected *i.e.,* $\mathcal{D} = \{(\mathbf{x}, y = f(\mathbf{x}) + \eta)\}$ where $\eta \sim \mathcal{N}(0, \sigma_\eta^2)$ is the Gaussian noise. The GP distribution ($\mathcal{GP}_f$) may be used to optimize $f$ using a BO algorithm with Thompson Sampling (TS) (Thompson, 1933) strategy.

---

[1]This makes the acquisition function less sensitive to uncertain augmentation features by reducing the rate of change/gradient with respect to that input.

At each iteration $t$, we compare the predictive likelihoods ($\mathcal{L}_t$) of both the human augmented arm and Control arm to select the model to use for suggesting the next promising candidate for the function evaluation. Then, we use Thompson sampling to draw a sample $S_t$ from the GP posterior distribution of the chosen arm and find its corresponding maxima:

$$\mathbf{x}_t^{\mathfrak{h}} = \operatorname*{argmax}_{\mathbf{x} \in \mathcal{X}} \left( S^{\mathfrak{h}}(\tilde{\mathbf{x}}) \right) \qquad \text{or} \qquad \mathbf{x}_t^{\mathfrak{f}} = \operatorname*{argmax}_{\mathbf{x} \in \mathcal{X}} \left( S^{\mathfrak{f}}(\mathbf{x}) \right)$$

Next, $f$ is observed at the suggested location *i.e.,* $(\mathbf{x}_t^{\mathfrak{h}}, f(\mathbf{x}_t^{\mathfrak{h}}))$ or $(\mathbf{x}_t^{\mathfrak{f}}, f(\mathbf{x}_t^{\mathfrak{f}}))$ and the rank GPs are updated to capture the preferences with respect to the new suggestion $\mathbf{x}_t^{\mathfrak{h}}$ or $\mathbf{x}_t^{\mathfrak{f}}$. Then, the main GPs modelling $f$ (both $\mathcal{GP}_f$ and $\mathcal{GP}_h$) are also updated accordingly. This process continues until the budget $T$ depletes. Additional details of BOAP framework is provided in the Appendix (§ A.3).

---

**Algorithm 1** Bayesian Optimization with Preferences on Abstract Properties (BOAP)

---

**Input**: Initial Observations: $\mathcal{D}_{1:t'} = \{\mathbf{x}_{1:t'}, \mathbf{y}_{1:t'}\}$, Preferences: $P^{\omega_i} = \{(\mathbf{x} \succ \mathbf{x}')_{1:p} \,|\, \forall i \in \mathbb{N}_m\}$

---

1. **for** $t = t' + 1, \cdots, T$ iterations **do**
2.      optimize hyperparameters $\Theta_t^{\star} = \{\theta_{\omega_{1:m}}^{\star}, \theta_h^{\star}, \theta_f^{\star}\}$ and update $\mathcal{GP}_{\omega_{1:m}}, \mathcal{GP}_h, \mathcal{GP}_f$
3.      compute predictive likelihoods $\mathcal{L}_t^{\mathfrak{h}}$ and $\mathcal{L}_t^{\mathfrak{f}}$ for Arm-$\mathfrak{h}$ and Arm-$\mathfrak{f}$
4.      **if** $\mathcal{L}_t^{\mathfrak{h}} > \mathcal{L}_t^{\mathfrak{f}}$, **then**
5.          draw a random sample $S_t^{\mathfrak{h}}$ from Arm-$\mathfrak{h}$ using Thompson Sampling
6.          maximize $S_t^{\mathfrak{h}}$ to find $\mathbf{x}_t^{\mathfrak{h}} = \operatorname*{argmax}_{\mathbf{x} \in \mathcal{X}} \left( S_t^{\mathfrak{h}}(\tilde{\mathbf{x}}) \right)$
7.          $\mathbf{x}_t = \mathbf{x}_t^{\mathfrak{h}}$
8.      **else**,
9.          draw a random sample $S_t^{\mathfrak{f}}$ Arm-$\mathfrak{f}$ using Thompson Sampling
10.          maximize $S_t^{\mathfrak{f}}$ to find $\mathbf{x}_t^{\mathfrak{f}} = \operatorname*{argmax}_{\mathbf{x} \in \mathcal{X}} \left( S_t^{\mathfrak{f}}(\mathbf{x}) \right)$
11.          $\mathbf{x}_t = \mathbf{x}_t^{\mathfrak{f}}$
12.      evaluate $f$ at $\mathbf{x}_t$ to obtain $y_t = f(\mathbf{x}_t) + \eta_t$
13.      Augment data $\mathcal{D} = \mathcal{D} \cup (\mathbf{x}_t, y_t)$ and update expert preferences $P^{\omega_{1:m}}$ with respect to $\mathbf{x}_t$
14.      $\mathbf{x}^{\star} = \operatorname*{argmax}_{(\mathbf{x}, y) \in \mathcal{D}} y$
15. **end for**
16. return $\mathbf{x}^{\star}$

---

### 3.4 Convergence Remarks

In this section we discuss the convergence properties of BOAP algorithm. The essence of BOAP is the combination of Thompson-Sampling Bayesian Optimization (TS-BO) with likelihood-based model-selection from multiple models of the same objective. To understand the convergence properties, we begin with special cases and then generalize to the real case.

**Purely Un-Augmented Case:** Let us suppose that the non-augmented (Control arm) model is used for all iterations of BOAP. In this mode of operation BOAP will operate precisely like standard Thompson-sampling BO (TS-BO), and thus regret is bounded as per TS-BO (Russo & Van Roy, 2014; Kandasamy et al., 2018; Chowdhury & Gopalan, 2017a) - i.e., in this simplified case the (cumulative) regret is bounded as $R_T \sim \mathcal{O}(\sqrt{T}(B\sqrt{\gamma_T} + \gamma_T))$ if we assume $f \in \mathcal{H}_k$ and $\|B\|_{\mathcal{H}_k} \leq B$, where $\gamma_T$ is the maximum information gain (which is controlled by the covariance prior $k$). Alternatively, assuming $f \in \mathcal{F}$, the Bayes regret is bounded as $\mathrm{BR}_T \sim \mathcal{O}(\sqrt{\dim_E(\mathcal{F}, \frac{1}{T})T \log T})$

(Russo & Van Roy, 2014), where $\dim_E(\mathcal{F}, \frac{1}{T})$ is the eluder dimension of $\mathcal{F}$ (Russo & Van Roy, 2014; Li et al., 2021) as defined below (see Definition 3.2).

**Definition 3.1 ($\epsilon$-independence)** *(Russo & Van Roy, 2014) Let $\mathcal{F} \subset \mathbb{R}^{\mathcal{X}}$, $\epsilon \in \mathbb{R}^+$. Then $\mathbf{x} \in \mathcal{X}$ is $\epsilon$-dependent of $\{\mathbf{x}_1, \mathbf{x}_2, \ldots, \mathbf{x}_n\} \subset \mathcal{X}$ if, $\forall f, f' \in \mathcal{F}$ such that $\sum_i (f(\mathbf{x}_i) - f'(\mathbf{x}_i))^2 \leq \epsilon^2$, then $|f(\mathbf{x}) - f'(\mathbf{x})| < \epsilon$. We say that $\mathbf{x} \in \mathcal{X}$ is $\epsilon$-independent if it is not $\epsilon$-dependent.*

**Definition 3.2 (Eluder dimension)** *(Russo & Van Roy, 2014) Let $\mathcal{F} \subset \mathbb{R}^{\mathcal{X}}$, $\epsilon \in \mathbb{R}^+$. The eluder dimension $\dim_E(\mathcal{F}, \epsilon)$ of $\mathcal{F}$ is the length of the longest sequence $\{\mathbf{x}_1, \mathbf{x}_2, \ldots, \mathbf{x}_n\} \subset \mathcal{X}$ such that $\forall i, \exists \epsilon' \geq \epsilon$ such that $\mathbf{x}_i$ is $\epsilon'$-independent of $\{\mathbf{x}_1, \mathbf{x}_2, \ldots, \mathbf{x}_{i-1}\}$.*

In a nutshell, eluder dimension is a measure of effective dimension, being the number of observations required to model any function in the set to within a given accuracy.

**Purely Augmented Case:** Next, suppose that the augmented model (Human arm) is used for all iterations of BOAP. In this case the overall objective model is not a GP (due to the involvement of rank GP as an input to the main GP), so we cannot naively apply information-gain based TS-BO regret bounds as we may for the Control arm case. However we can apply the eluder dimension bounds. Whether this provides better convergence depends entirely on the relevance and accuracy of the user expert feedback used to construct the model augmentation: accurate feedback of relevant abstract properties should, we postulate, reduce the eluder dimension of the model (with the limiting case where the augmentation models $f$), while inaccurate or irrelevant feedback may mislead the model and increase eluder dimension, impeding convergence.

**General Case:** More generally, BOAP may select between the Control and Human Arms, which can be modeled stochastically. If neither model has a consistently higher likelihood (either initially or asymptotically) then the convergence behaviour will follow the worst-case convergence of TS-BO using either the un-augmented or augmented model alone - that is:

$$\text{BR}_T \sim \mathcal{O}(\sqrt{\max\{\dim_E(\mathcal{F}_\natural, \tfrac{1}{T}), \dim_E(\mathcal{F}_\flat, \tfrac{1}{T})\}T\log T})$$

Under the reasonable assumptions that the predictive likelihood is (a) an accurate measure of model fit, and that asymptotically (b) $\mathcal{L}_t^\natural > \mathcal{L}_t^\flat$ if $\dim_E(\mathcal{F}_\natural, \tfrac{1}{T}) < \dim_E(\mathcal{F}_\flat, \tfrac{1}{T})$ (and vice-versa) then:

$$\text{BR}_T \sim \mathcal{O}(\sqrt{\min\{\dim_E(\mathcal{F}_\natural, \tfrac{1}{T}), \dim_E(\mathcal{F}_\flat, \tfrac{1}{T})\}T\log T})$$

So, provided that the expert provides accurate and relevant preference feedback we would have that (a) the augmented model will (asymptotically) be selected and thus dominate the regret bound, and (b) due to its lower eluder dimension the regret bound will be tighter, leading to a faster convergence.

## 4 EXPERIMENTS

We evaluate the performance of BOAP method using synthetic benchmark function optimization problems and real-world optimization problems arising in advanced battery manufacturing processes. We have considered the following experimental settings for BOAP. We use the popular Automatic Relevance Determination (ARD) kernel (Neal, 2012) for the construction of both the rank GPs and the conventional (un-augmented) GPs. For rank GPs, we tune ARD kernel hyperparameters $\theta_d = \{l_d\}$ using max-likelihood estimation (Eq. (5)). For the augmented GP modeling $\tilde{\mathbf{x}}$, we use a modified ARD kernel that uses spatially varying kernel with a parametric lengthscale function (See discussion in § 3.2). As we normalize the bounds, we tune $l_d$ (the lengthscale for the *un*-augmented features) in the interval $[0.1, 1]$ and the scale parameter $\alpha$ (for the auxiliary features) in the interval $(0, 2]$. Further, we set signal variance $\sigma_f^2 = 1$ as we standardize the outputs.

We compare the performance of BOAP algorithm with the following state-of-the-art baselines. **(i) BO-TS:** a standard Bayesian Optimization (BO) with Thompson Sampling (TS) strategy, **(ii) BO-EI:** BO with Expected Improvement (EI) acquisition function, and **(iii) BOAP - Only Augmentation (BOAP-OA)**: In this baseline, we run our algorithm without the 2-arm scheme and we only use augmented input for GP modeling. This baseline shows the effectiveness of expert's inputs. We do not consider any preference based BO methods González et al. (2017); Benavoli et al. (2021) as baselines, because the preferences are provided directly on the objective function, as opposed to abstract properties that are not measured directly. The additional details of our experimental setup are provided in the Appendix (§ A.4.1).

Table 1: Details of the synthetic optimization benchmark functions. Analytical forms are provided in the 2$^{nd}$ column and the last column depicts the high level features used by a simulated expert.

| Functions | $f(\mathbf{x})$ | High Level Features |
|---|---|---|
| Benchmark-1d | $\exp^{(2-\mathbf{x})^2} + \exp^{\frac{(6-\mathbf{x})^2}{10}} + \frac{1}{\mathbf{x}^2+1}$ | $\omega_1 = \exp^{(2-\mathbf{x})^2}, \omega_2 = \frac{1}{\mathbf{x}^2}$ |
| Rosenbrock-3d | $\sum\limits_{i=1}^{d-1}[100 \times (x_{i+1} - x_i^2)^2 + (x_i - 1)^2]$ | $\omega_1 = (x_3 - x_2^2)^2 + (x_2 - x_1^2)^2$ $\omega_2 = (x_2 - 1)^2 + (x_1 - 1)^2$ |
| Griewank-5d | $\sum\limits_{i=1}^{d}\left[\frac{x_i^2}{4000} - \prod\limits_{i=1}^{d}\cos\left(\frac{x_i}{\sqrt{i}}\right) + 1\right]$ | $\omega_1 = \sum\limits_{i=1}^{d} x_i^2, \ \omega_2 = \prod\limits_{i=1}^{d}\cos x_i$ |

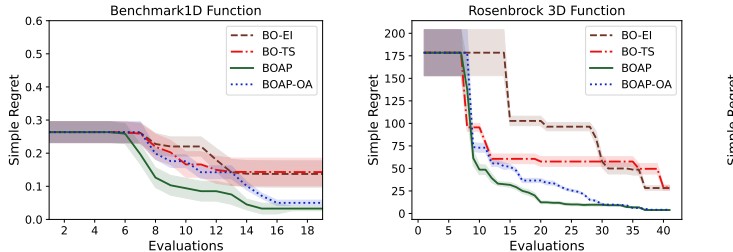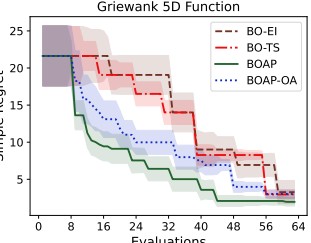

Figure 2: Simple regret vs iterations for the synthetic multi-dimensional benchmark functions. We plot the average regret along with its standard error obtained after 10 random repeated runs.

## 4.1 SYNTHETIC EXPERIMENTS

We evaluate BOAP framework in the global optimization of synthetic optimization benchmark functions (Surjanovic & Bingham, 2017). The list of synthetic functions used are provided in Table 1.

**Emulating Expert's Preferential Inputs:** As discussed in §3.1, we fit a rank GP using the expert preferences provided on designs based on their cognitive knowledge. In all our synthetic experiments we set $m = 2$, *i.e.,* we model two abstract properties $\{\omega_1, \omega_2\}$ for the considered synthetic function. We expect the expert to know the higher order abstract features of each design $\mathbf{x} \in \mathcal{X}$. We construct rank GPs by emulating the expert preferences based on such high-level features of the given synthetic function. The possible set of high-level features of the synthetic functions are mentioned in Table 1. We generate preference list $P^{\omega_{1:2}}$ for each high-level feature of the designs by comparing its utility. We start with $p = \binom{t'}{2}$ preferences in $P$, that gets updated in every iteration of the optimization process. We construct rank GP surrogates $\{\mathcal{GP}_{\omega_1}, \mathcal{GP}_{\omega_2}\}$ using $P^{\omega_1}$ and $P^{\omega_2}$.

For a given $d-$dimensional problem, we have considered $t' = d+3$ initial observations and allocate the overall budget $T = 10 \times d + 5$. We repeat all our synthetic experiments 10 times with random initialization and report the average simple regret (Brochu et al., 2010) as a function of iterations. The convergence plots obtained for the global optimization experiments using synthetic functions after 10 runs are shown in Figure 2. It is evident from the convergence results that our proposed BOAP method has outperformed the standard baselines by a huge margin, thereby proving its superiority.

To demonstrate the effectiveness of BOAP we have conducted additional experiments by accounting for the inaccuracy or poor choices in expert preferential knowledge. Due to the space constraints the ablation results and the additional experimental results are provided in the Appendix (§ A.4).

## 4.2 REAL-WORLD EXPERIMENTS

We compare the performance of BOAP in two real-world optimization use-cases in Lithium-ion battery manufacturing that are proven to be very complex and expensive in nature, thus providing a wide scope for the optimization. Further, battery scientists often reveal additional knowledge about the abstract properties in the battery design space and thus providing a rich playground for the evaluation of our framework. We refer to the appendix (§ A.4.3) for the detailed experimental setup.

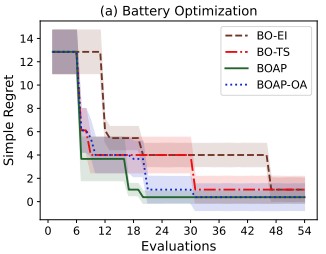
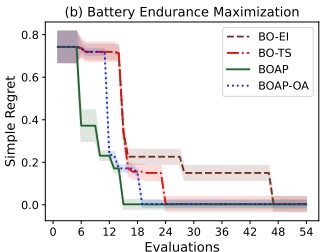

Figure 3: Simple regret vs iterations for battery manufacturing optimization experiments: **(a)** Optimization of electrode calendering process, and **(b)** Optimization of battery endurance.

### 4.2.1 OPTIMIZATION OF ELECTRODE CALENDERING

In this experiment, we consider a case study on the calendering process proposed by Duquesnoy et al. (2020). The authors analyzed the effect of parameters such as calendering pressure ($\varepsilon_{cal}$), electrode porosity and electrode composition on the electrode properties such as electrolyte conductivity, tortuosity (both in solid phase ($\tau_{sol}$) and liquid phase ($\tau_{liq}$)), Current Collector (CC), Active Surface (AS), etc. We define an optimization paradigm using the data published by Duquesnoy et al. (2020).

We use our proposed BOAP framework to optimize the electrode calendering process by maximizing the *Active Surface* of electrodes by modeling two abstract properties: **(i)** Property 1 ($\omega_1$): *Tortuosity in liquid phase* $\tau_{liq}$, and **(ii)** Property 2 ($\omega_2$): *Output Porosity* (OP). We simulate the expert pairwise preferential inputs $\{P^{\omega_{\tau_{liq}}}, P^{\omega_{OP}}\}$ by comparing the actual measurements reported in the dataset published by Duquesnoy et al. (2020). We consider 4 initial observations and maximize the active surface of the electrodes for 50 iterations. We compare the performance of our proposed BOAP framework by plotting the average simple regret after 10 repeated runs with random initializations. The convergence results obtained for the electrode calendering optimization is shown in Figure 3a.

### 4.2.2 ELECTRODE MANUFACTURING OPTIMIZATION

The best battery formulation and the optimal selection of process parameters is crucial for manufacturing long-life and energy-dense batteries. Drakopoulos et al. (2021) analyzed the manufacturing of Lithium-ion graphite based electrodes and reported the process parameters in manufacturing a battery along with the output charge capacities of the battery measured after certain charge-discharge cycles. In our experiment, we use BOAP to optimally select the manufacturing process parameters to design a battery with maximum endurance *i.e,* a battery that can retain the maximum charge after certain charge-discharge cycles. We consider *Anode Thickness* (AT) and *Active Mass* (AM) as abstract properties $\{\omega_{AT}, \omega_{AM}\}$ to maximize the battery endurance $E = \frac{D_{50}}{D_5}$, where $D_{50}$ and $D_5$ are the discharge capacities of the cell at $50^{th}$ and $5^{th}$ cycle, respectively. We consider 4 initial observations and maximize the endurance of the cell for 50 iterations. We compare the performance by plotting the average simple regret versus iterations after 10 random repeated runs. The convergence results obtained for maximizing the battery endurance is shown in Figure 3b.

It is evident from Figure 3 that BOAP is superior to the baselines due to its ability to model the abstract properties of the battery designs that can be beneficial in accelerating BO performance.

## 5 CONCLUSION

We present a novel approach for human-AI collaborative BO for modeling the expert inputs on abstract properties to further accelerate the sample-efficiency of BO. Experts provide preferential inputs about the abstract and unmeasurable properties. We model such preferential inputs using rank GPs. We augment the inputs of a standard GP with the output of such auxiliary rank GPs to learn the underlying preferences in the instance space. We use a 2-arm strategy to overcome any futile expert bias and encourage exploration. We discuss the convergence of our proposed BOAP framework. The experimental results show the superiority of our proposed BOAP algorithm.

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
