# OpenReview forum: "Enhanced Bayesian Optimization via Preferential Modeling of Abstract Properties"
_ICLR.cc/2024/Conference — Submitted to ICLR 2024_

### Official Review · Reviewer_hhJX · 2023-10-19

**Soundness:** 3 good
**Presentation:** 3 good
**Contribution:** 2 fair
**Rating:** 5
**Confidence:** 4

**Summary:**

The authors propose to ease black-box optimization tasks by integrating both black-box function evaluations over input designs as well as human preferences between different designs. In the Bayesian framework, this boils down to a combination between Bayesian Optimization (BO) and Preferential BO (PBO). Human preferences can be of significant interest as 1) they are cheaper with respect to black-box function evaluations for costly objectives and 2) humans perform comparisons at an abstract level using concepts that may not be easily measurable. The proposed method, Bayesian Optimization with Abstract Properties (BOAP), integrates human preferences directly into the statistical surrogate used during BO by extending the input space with additional dimensions that contain learned human preferences. To further account for unreliable human feedback, BOAP keeps track of two surrogates, with and without human preferences, and selects which one should be used to obtain a new design at every iteration based on a criterion.
 BOAP is then evaluated on a range of problems and demonstrates an improvement over the vanilla BO setting, even for unreliable experts.

**Strengths:**

- The method is easy to understand and well-described throughout the paper. It provides a simple way to leverage different kinds of data (scalar values of a black box function and human preferences).
- The authors give some hints from a theoretical perspective as to why using expert feedback might lead to faster convergence.

**Weaknesses:**

- There is no notion of human query cost compared to the black-box function evaluation cost. For expensive black-box functions, I can easily imagine that querying the human is cheaper, but if many comparisons are needed. However, as such, it is difficult to assess the benefit of the method. BOAP provides a way to integrate expert preferences, and yes this yields an improvement for relevant preferences, but this is assuming a null human query cost, and perhaps things would be different
This is even more salient for unreliable expert feedback. For instance, Figure 6 shows the performances of the method for noisy expert preferences. While it seems that BOAP is somehow robust to noisy expert feedback, reverting back to vanilla BO in the worst case, it most likely would be the worst competitor if the expert query cost is now added.
This being said, this "weakness" is directed towards the evaluation protocol rather than the method itself.

This is more of a remark than a weakness properly speaking: I believe that the ICLR conference is generally skewed towards Deep Learning methods and the many approaches that revolve around that field. Bayesian Optimization has been quite successful for hyperparameter optimization of complex deep learning models and therefore fits the scope. However, the present submission does not seem to aim for such applications. In particular, no deep learning-related applications were provided in the experiments. Furthermore, as this submission proposes a way to integrate human preferences to the BO setting, I would say that comparing hyperparameter sets is probably not the most manageable problem for human experts, hence I think that the submission may not wholly fit the scope of ICLR.

**Questions:**

Q1: During experiments, we begin with $p = (t' 2)$ preferences, ``that gets updated in every iteration of the optimization process''. Can you clarify this? If we begin with 5 initial observations, there are 10 preferences. Once the 6th sample is acquired, is it compared with the 5 previous ones, so that we now have 15 preferences, and 21 in the next iteration, and so on? If so, this leads to a great number of comparisons, which again raises the question of the human query cost w.r.t. black-box function evaluation cost mentioned above.

Q2: In all the experiments considered, the expert is simulated, and considered to reason over high-level features which are then used to perform the comparisons and learn the preferential Gaussian process whose posterior mean is then added as additional input dimensions to the BO surrogate. How would this method compare with actually using the high-level features directly as additional dimensions? e.g. for Benchmark-1d, the input space would be $[x, \exp(2-x)^2, \frac{1}{x^2}]$. I think this can be intriguing specifically given that preferences are only ''identifiable'' up to a monotonous transformation (e.g. if an expert would have latent utility function $x \mapsto x$ or $x \mapsto \sqrt{x}$, he would still give the same answer for a given comparison). If performances are significantly different, one could investigate whether this has to do with this identifiability issue or not. This can also be done in "real-world experiments" as high-level features are also derived in this case.

Q3: P5 of the appendix, it is mentioned that ``if we use all training instances for the computation of the log marginal likelihood, there are chances that only Control arm may get selected in majority of the rounds. Therefore, to avoid this, instead of using all the training instances for computing the marginal likelihood, we use only the subset of the original training data for finding the optimal hyperparameter set and then we use the held-out instances from the original training set to compute the (predictive) likelihood''. I must say that I don't fully understand this phenomenon, could you clarify why this happens?

Q4: To select which GP surrogate to follow (with or without human preferences and additional dimensions), the predictive likelihood of both models is compared. But the latter does not incorporate a notion of model complexity, right? I would have thought that the augmented model better fits the data given that it has more flexibility, even though for Gaussian Processes in a low-data regime (as in BO), having extra dimensions might not be a blessing.
From a more general perspective, I think that mentioning additional ways of selecting which surrogate to use would have benefited the paper. I can think of at least one: placing a sparsifying-prior on the (squared-inverse) lengthscales, such as the horseshoe prior, as was done by Eriksson and Jankowiak [1].

Some additional remarks:

- Simple regret and Bayes Regret are never defined formally in the paper, adding a definition in supplementary would be valuable.
- On the contrary, the UCB acquisition function is defined in the supplementary, but nowhere used in the paper.
- In the Supplementary, Eq.17 involves two different notations for a function $\Gamma$
- I don't think the search space for the synthetic experiments is mentioned anywhere in the text.

[1] High-Dimensional Bayesian Optimization with Sparse Axis-Aligned Subspaces, David Eriksson, Martin Jankowiak, https://arxiv.org/abs/2103.00349

---

> ### Author Response · Authors · 2023-11-22
> **Response to reviewer hhJX**
>
> We thank our reviewer's time and effort for providing a detailed review and raising interesting key points.
>
> - **BOAP assumes null human query cost**_
>     - Bayesian optimization (BO) is the workhorse of scientific experimentation and product design. It provides a rich platform for human-AI collaboration because the nature of BO applications ensures that there would be an expert running the optimization process. In BOAP, the expert inputs considered are qualitative and available as pairwise design preferences
> based on some abstract properties. Eliciting such pairwise rankings from the human expert should not add significant overhead, in contrast to specifying the complex explicit knowledge.
>     -  BOAP allows an expert to intervene only when they are willing to provide any preferential input thereby allowing them to evolve their knowledge over the course of optimization. Therefore, we have not considered any human query costs in the current version of the manuscript. However, it is an interesting direction to pursue that makes our proposed BOAP framework even more robust. In our experiments, at each iteration, we randomly choose $n/2$ datapoints to generate preferential data with the newly suggested datapoint.
>
> - _**Formal definitions: Bayes regret, simple regret; Search space for the synthetic experiments**_
>     - We thank our reviewer for the suggestions, and we will provide them in the updated version of the manuscript.
>
> - **How would this method compare with actually using the high-level features directly as additional dimensions?**_
>     - We would like to clarify to our reviewers that high-level features are not known directly to be used as an additional dimension, instead in practice an expert would reason about a system only qualitatively. However, we have conducted additional experiments to directly use high-level features as additional dimensions (BOAP-AD) and we have found that it outperforms BOAP by a small margin we believe this phenomenon is due to the fact that explicit measurements are always better than approximate qualitative measurements.
>
>      -----------------------------------------------------
>       |                 |      BOAP-AD     |       BOAP     |
>      -----------------------------------------------------
>       | Benchmark - 1D  |     0.01±0.002   |     0.05±0.01  |
>       | Rosenbrock - 3D |     3.16±1.19    |     5.32±0.12  |
>      -----------------------------------------------------

---

### Official Review · Reviewer_cKDa · 2023-10-29

**Soundness:** 1 poor
**Presentation:** 2 fair
**Contribution:** 2 fair
**Rating:** 1
**Confidence:** 4

**Summary:**

The paper proposes an approach for incorporating abstract properties, a novel form of prior information, into of the objective function into the Bayesian optimization loop. Abstract properties refer to auxiliary, immeasurable traits of the objective, which by assumption are indicative of performance. User preferences over abstract properties are modeled by a rank GP. The rankings are subsequently incorporated by adding dimensions to a conventional GP regression that is subsequently used for BO.

**Strengths:**

__Interesting idea:__ Querying the user for feedback (specifically for ranking-based feedback) appears useful and digestible for practitioners.

__Novel quantities:__ I have not seen abstract properties be discussed as a concept previously, and while I am not convinced of their prevalence in practical applications, the notion of incorporating _immeasurable_ auxiliary quantities is enticing.

**Weaknesses:**

Unfortunately, I believe this paper has substantial flaws in terms of the validity of the method, the presented theory, results and communication. As such, I believe this paper needs substantial re-work for it to be publishable.


- __Convergence remarks:__ This subsection is rather informal. I do not believe that postulation is appropriate when considering theoretical convergence, but the assumption also appears incorrect based on existing theory.
_(...) accurate feedback of relevant abstract properties should, we postulate, reduce the eluder dimension of the model_.

Regarding convergence of the proposed method, I believe there is only one certainty: the proposed method _adds_ dimensions to the problem. As shown by Srinivas. et. al. (2009), the information gain scales exponentially in the dimensionality of the problem, assuming equal lengthscales (Bergenkamp et. al. 2017). Furthermore, the eluder dimension clearly increases in the dimensionality of the problem ($\mathcal{F}$ changes with the added dimensions) so the above statement is, to the best our knowledge, incorrect.

Lastly, the bounds on _information gain_ (IG) only hold for a select few kernels, and spatially-varying kernels are not included in this class. As such, IG bounds do not hold, either.

The entirety of Sec. 3.4 is based on seemingly incorrect assumptions. Further, it contains no _proven_ theoretical insights. As such, I believe this section should be fundamentally reworked or removed.

- __Addition of dimenions:__ I struggle to see how adding dimensions to the problem aids in optimization. This is a counterintuitive action, and the authors should undertake great effort to shed light on this. This is currently missing from the paper altogether.

- __Obscure aspects of method:__ I believe a collection of methodological choices are not sufficiently communicated. These are all outlined in the questions section.

- __The concept of abstract properties:__ Moreover, the abstract properties that are introduced appear measurable, and as such, amenable to multi-objective (or multi-fidelity, depending on the property) optimization. Since these properties are _abstract_, I would encourage the authors to further motivate why they are abstract (i.e. immeasurable) and why existing BO approaches are ill-suited.

- __Structure of the methodology section:__ A substantial part of this section (3.1.1, 3.1.2) appears to be background work, and 3.1.2 (MAP estimation) specifically is standard BO convention. The methodology should primarily cover the novel aspects of the work, so including these here blurs the contributions of the work. I suggest for 3.1.1. to be moved to Sec. 2 and 3.1.2 to be moved to the Appendix. The same goes for ARD (3.2 second paragraph).

- __Notation:__ The set $\mathcal{D} = \{(\mathbf{x}, y = h(\mathbf{\hat{x}}) \approx f(\mathbf{x})\})$ is incorrect notation and difficult to parse regardless. Moreover, observations are occasionally denoted $(\mathbf{x}, f(\mathbf{x}))$, should be $(\mathbf{x}, y)$.
- __Results:__ Few tasks, very few iterations and low repetitions (as seen by the very non-smooth regret plots) yield results that have low credibility and offer few substantial takeaways. Moreover, the high-level features seem _very helpful_, so I believe these should be substantially ablated.


__Minor:__
- Variables are re-declared multiple times throughout
- The rank GP is frequently referred to as _"Rank (preferential) GP"_. Please use either rank or preferential, as the dual naming and parentheses do not add clarity.
- _Information gain_ versus _information-gain_. Please use the former.

**Questions:**

__Methodology-related questions - should be clarified:__
- Why are the lengthscales fixed (as opposed to the conventional ARD) for the original input dimensions? How are they fixed?
- What is the _"maximum predictive likelihood"_ by which the model is chosen -  the one with the highest marginal log likelihood?
- One model strictly has more capacity than the other, so how would the non-user input model ever be chosen?
 - How frequently is each model chosen?
- How is the next input chosen along the added $m$ dimensions, which are dictated by human feedback?
- Are the $m$ dimensions amenable to typical acquisition function optimization, and do they actually result in a point to query?  (it seems like _output_, rather than _input_)

__General questions__:
- What is the procedure for providing abstract features in the results? Can these be provided as continous functions, or are they given pointwise? If pointwise, how are they provided for the synthetic functions?
- What is the definition of an abstract property? Immeasurable? If so, I would suggest calling it "immeasurable", as calling it "abstract" is, quite fittingly, a bit abstract.
- How frequently does the user have to be queried for feedback / how many user queries does a 20 iteration run entail?

---

> ### Author Response · Authors · 2023-11-22
> **Response to reviewer cKDa**
>
> We appreciate our reviewer's patience, time and effort to provide us with detailed feedback and raise interesting key points.
>
> - _**Why properties are abstract (i.e. immeasurable) and why existing BO approaches are ill-suited**_
>     - In numerous scenarios, experts often reason about a system based on their approximate or qualitative knowledge of the system at different abstraction levels. They reason using intermediate physical properties that may not be measurable directly. Using such high-level abstractions the expert compares designs and the reason why a design is better than another.
>     - The existing BO approaches aim at capturing the expert preferential input directly on the objective function, in contrast to the preferences based on the aforesaid intermediate properties. Therefore we propose a human-AI collaborative framework to incorporate such expert preferential data about unmeasured abstract properties into BO.
>
> - _**Why are the lengthscales fixed for the original input dimensions?**_
>     - We would like to clarify to our reviewer that in the augmented GP ($\mathcal{GP}_{h}$), we still tune the hyperparameters of the original input dimensions by maximizing the log marginal likelihood. We use a modified ARD kernel that uses a spatially varying kernel with a parametric lengthscale function for each of the input dimensions. For each un-augmented feature, we set the lengthscale function to be a constant lengthscale value $l(\mathbf{x})=l$, whereas for each of the augmented auxiliary features we set the lengthscale function to be $l(x)=\alpha\sigma(\mathbf{x})$, where $\alpha$ is the scale parameter and $\sigma(\mathbf{x})$ is the normalized standard deviation predicted using the rank GP.
>
> - _**How would the non-user input model ever be chosen? How frequently it is chosen?**_
>
>     - During the initial phase of optimization the un-augmented control arm is pulled as the augmented arm may have insufficient data to reduce uncertainty in posteriors, making the augmented features only slightly better than noise. However, the augmented features become more accurate through the optimization process as we accumulate more data, so we observe the augmented arm to be more accurate at the end of the optimization.
>     - Based on the empirical results we could verify that the augmented model was chosen more than the un-augmented arm. In an ideal scenario with accurate preferential data, we have recorded the percentage of times each was pulled and the results are as follows.
>      -----------------------------------------------------
>       |            |   Augmented Arm   |   Control Arm   |
>      -----------------------------------------------------
>       | Benchmark  |     76.23±2.5     |     24.22±1.2   |
>       | Rosenbrock |     71.87±4.6     |     25.67±2.75  |
>       | Griewank   |     63.95±3.1     |     35.73±2.6   |
>
>
> - _**{How is the next input chosen along the added dimensions?**_
>     - In the augmented search space we have $d+m$ features in total, here we Thompson sampling method operating in $d+m$ dimensions to find the next promising candidate for the function evaluation. In the case when the augmented arm is pulled, we randomly draw a sample from its GP and find its corresponding maxima $\mathbf{x}_{t}^{\mathfrak{h}}$.
> The objective function in the augmented space $h(\tilde{\mathbf{x}})$ is a simpler form of $f(\mathbf{x})$ with auxiliary features in the
> input, and therefore we observe $h(\tilde{\mathbf{x}})$ via $f(\mathbf{x})$.
>
> - _**How frequently does the user have to be queried for feedback, and how many user queries do a 20-iteration run entail.**_
>     - BOAP allows an expert to intervene only when they are willing to provide any preferential input thereby allowing them to evolve their knowledge over the course of optimization. The preferential feedback is provided by experts at their discretion i.e., when they feel they can make a clear judgement or that the algorithm has gone astray. How the expert intervenes in giving preferential feedback is not controlled by the algorithm.
>     - In our studies with simulated experts (zero query cost), at each iteration, we randomly choose a subset of datapoints containing $n/2$ datapoints to compare with the new candidate suggested for function evaluation. Therefore in a $20$ iteration run, we generate $133$ preferences in total.
>
> - _**Procedure for providing abstract features**_
>     - For synthetic functions, based on their mathematical formulation, we have considered the high-level features as mentioned in Table 1 of the main paper. We treat such high-level features as a continuous function $\omega(\cdot$) and generate the preference data $P$. For real-world experiments, due to the cost and access limitations of experimental facilities we have used a few columns $(\omega)$ in the published dataset as high-level features. We generate preference data by comparing those columns values $(\omega(\mathbf{x}))$ for two datapoints $(\mathbf{x},\mathbf{x}')$.

---

> ### Comment · Reviewer_cKDa · 2023-12-05
> **Follow-up**
>
> Thanks to the authors for their clarification.
>
> __Why are the lengthscales fixed for the original input dimensions?__
> These, to me, still seem like unconventional choices. I would encourage the authors to carefully motivate why some lengthscales are fixed (which is a rather drastic modeling choice) so that the proposed method can be evaluated on its own merit.
>
>  __How is the next input chosen along the added dimensions?__
>
> I am still not quite following. Along the preferences, conducting TS, to me, seems to equate to _choosing a preference_ (which are determined by the user, not by BO). As such, I am not following how this can be done in practice.
>
> __How would the non-user input model ever be chosen? How frequently it is chosen?__
>
> I do not believe the right question was answered here. My question, specifically, is:
> Since the augmented model has a larger capacity (more dimensions), it should in theory yield a larger likelihood at all times if they are compared on the same (subset of the) data. Yet, it does not seem to do so. Why does it not have a larger likelihood at all times? Surely, I must be missing some aspect of the algorithm here, and it would be great if the authors could clarify what this is.
>
> __How frequently does the user have to be queried for feedback, and how many user queries do a 20-iteration run entail?__
> This is valuable context, and I believe its inclusion in the paper would strengthen it.
>
>
>
> My concerns regarding the theoretical discussion have unfortunately not been addressed. My overall perception of the paper has not improved, and as such, I will retain my score.

---

### Official Review · Reviewer_4RNk · 2023-11-01

**Soundness:** 2 fair
**Presentation:** 2 fair
**Contribution:** 2 fair
**Rating:** 3
**Confidence:** 4

**Summary:**

This paper proposes novel human-AI collaboration BO algorithm: Bayesian Optimization with Abstract Properties (BOAP) where the input of the main GP f in BO is augmented with latent preference GPs posterior means to construct an augmented GP h. BOAP then adaptively switch between f and h when performing BO. The author examined BOAP via theoretical discussion and empirical evaluation.

**Strengths:**

- Incorporating human feedback into BO is an interesting and important problem.
- The proposed method is well-motivated as it seeks to incorporate human feedback.
- The paper has contained both theoretical and empirical evaluation, with the experimental results consist of both various synthetic and real-world problems

**Weaknesses:**

While this work presented a well-motivated method, many design choices throughout the algorithm appears to be arbitrary, and the author did not provide enough theoretical or experimental justification to those choices. Some examples include:

- (in 3.1) “To do this, we utilize an Automatic Relevance Determination (ARD) kernel where we set the lengthscales of the augmented input dimensions in proportion to the rank GP uncertainties” —> why this instead of learning the parameters as one would typically do with GP models? Have we run ablation studies on how exactly we are choosing the scale parameter \alpha? inverse std? inverse var? or some other notion of uncertainty? The author has provided some explanation to that choice, but it is not convincing that this is the best choice here without additional evidence.
- If we are able to model those abstract properties with preference models that we believe are compositional parts (or contain important information) of the function of interest f, why don’t we explicitly exploit this compositional structure as done in Astudillo and Frazier (2020)?
- Similar to above points, the author does not describe how the preference P is constructed and updated (i.e., how are the queries selected. Randomly?). Does P gets updated? If yes, how (L13 in algorithm 1 is unclear) If not then the number of possible comparisons entirely depend on the initial dataset, and are either very limited or we need a large init set, which isn’t feasible.?
- Theoretical discussion seems a bit hand-wavy and are based not entirely justified assumptions (e.g., “If neither model has a consistently higher likelihood…”)

The experiments section can also be significantly improved by exploring

- The paper would benefit more from investigating the impact of preference data P. Querying human experts is an expensive procedure, and in the BOAP algorithm, we have to query human expert p times for each of the
- Human mistakes are mentioned multiple times throughout the paper but the author doesn’t investigate how different kinds/scales of human errors can affect the performance of BOAP. Similarly, different human preference querying strategies are not explored. Astudillo and Frazier (2020) and Lin et al. (2022) have shown that querying the DM different questions (e.g., using the EUBO acquisition function) can have significant impact on the downstream model performance.

The writing, particularly implementation and design choices details (e.g., how are expert preferences dataset being constructed precisely), can be improved.

Finally, while the author has experimented with different test functions, there are only two real-world problems and the performance of BOAP in those problems are only marginally better than the baseline methods.

Minor points:

- L12-13 should be indented to be inside the loop.

References:

Astudillo, R., & Frazier, P. (2019, May). Bayesian optimization of composite functions. In *International Conference on Machine Learning* (pp. 354-363). PMLR.

Astudillo, R., & Frazier, P. (2020, June). Multi-attribute Bayesian optimization with interactive preference learning. In International Conference on Artificial Intelligence and Statistics (pp. 4496-4507). PMLR.

Lin, Z. J., Astudillo, R., Frazier, P., & Bakshy, E. (2022, May). Preference exploration for efficient bayesian optimization with multiple outcomes. In *International Conference on Artificial Intelligence and Statistics* (pp. 4235-4258). PMLR.

**Questions:**

- “Although we anticipate that experts will provide accurate preferences on abstract properties, the expert preferential knowledge can sometimes be misleading” —> what does misleading mean here?
- Have the author compared the impact of using log-likelihood of rank GP instead of Evidence, as the former appears not to take uncertainty in \omega into consideration, where the latter marginalize out the (latent) GP function distribution, arguably a more principled Bayesian treatment.
- (in 3.2) “To handle different scaling levels in rank GPs, we normalize its output in the interval [0, 1], such that μ(ω_i (x)) ∈ [0, 1]” → how is the normalization being done
- (Algorithm 1):"Augment data D = D \union (x_t ,y_t ) and update expert preferences Pω 1:m with respect to x_t” How exactly are the expert preferences data updated? How are pairwise comparisons being constructed?
- Other questions mentioned in Weakness

---

> ### Author Response · Authors · 2023-11-22
> **Response to reviewer 4RNk**
>
> We appreciate the reviewer's time and efforts to provide detailed feedback and raise a few interesting points.
>
> - _**Why this instead of learning the parameters as one would typically do with GP models?**_
>      - Tuning the human augmented GP ($\mathcal{GP}_{h}$) using the traditional loglikelihood/evidence maximization does not account for the expert uncertainties for the augmented features. Incorporating expert uncertainties from the individual rank GPs can be crucial for modelling as the expert feedback may be inaccurate and thus can adversely affect the overall optimization performance.
>
> - _**why don't we explicitly exploit this compositional structure as done in Astudillo and Frazier (2020)?**_
>     - Our idea is fundamentally different from Astudillo and Frazier (2020) as we try to incorporate expert preferential feedback about the immeasurable abstract properties in modelling the given objective function $f$. To the best of our knowledge, Astudillo and Frazier (2020) capture expert preferences (along with their uncertainties) directly on $f$ and fail to account for auxiliary abstract properties that can be crucial for improving the optimization performance.
>
>
> - _**The choice of scale parameter $\alpha$**_
>     - The predictive variance ($\sigma^{2}$) in the rank GP is a measure of uncertainty in the expert model. In our proposed approach, the scale at which this uncertainty is to be considered in modelling the overall objective function is dictated by the scale parameter $\alpha$ and $\alpha\times\sigma$ will be the underlying lengthscale. We tune the scale parameter for each of the augmented dimensions by maximizing the log-likelihood. Alternatively, we could treat $\alpha$ as an overall "average" lengthscale, such that $\alpha$ is the underlying lengthscale when there is no uncertainty and with expert uncertainty the overall lengthscale is $\alpha+\sigma$.
>
> - _**Different kinds of human errors can affect the performance of BOAP}**_
>     - We would like to refer our reviewer to the additional experiments mentioned in the supplementary material (Section A.4). We have conducted experiments to account for the human expert errors in providing preferential feedback. We have done this ablation study in two-fold. First, we show the robustness of our proposed approach against human errors in the selection of higher-order abstract properties. Second, we account for the human errors in providing the expert preferential feedback by flipping the preference between two input designs with some probability $\delta=0.3$. The ablation results and the experimental details are provided in the supplementary material (A.4.2).
>
>
> - _**Expert preferential knowledge can sometimes be misleading**_
>     - We expect that the human expert preferential input is accurate and thus likely to accelerate BO. But, in some cases, this preferential feedback may be inaccurate due to the wrong/incorrect assumptions on the given search space. Therefore, such misguided preferential data given by an expert could potentially slow down the optimization. On the other hand, our method has some immunity to such inaccurate feedback.
>
> - _**Normalization in Rank GPs**_
>     - The individual rank GPs (${\mathcal{GP_{\omega_i}}}$) operate at different output scales (as dictated by MAP) and thus different scaling levels in their corresponding predictions. Therefore, before we augment the main GP with the mean predictions ($\mu_{\omega_{i}}(\mathbf{x})$) of the individual rank GPs, we min-max scale their outputs to be normalized in the interval $[0,1]$.

---

### Official Review · Reviewer_NPJR · 2023-11-03

**Soundness:** 2 fair
**Presentation:** 3 good
**Contribution:** 2 fair
**Rating:** 3
**Confidence:** 3

**Summary:**

This paper explores a new Bayes Opt method that incorporates human rank-based feedback as well as direct experimental feedback. As such, it is a human-in-the-loop algorithm. To achieve this, multiple GP models are trained. For the human feedback, $m$ rank GPs are trained. The mean values of each rank GP are then incorporated as additional predictor variables in the main GP that predicts the overall objective. The uncertainty in the rank GPs are used to guide the lengthscales in the kernel of the main GP. Finally, a GP trained without any of the human-derived extra features is also trained, in case expert feedback turns out to be inaccurate. Experiments conducted: synthetic experiments, real dataset experiments where some data fields are regarded as expert derived ranks instead.

**Strengths:**

- The paper clearly describes the scenario that it is designed for
- The authors have attempted to use the uncertainty from the rank GPs in the main GP by incorporating the uncertainties into the lengthscale
- Some theoretical discussion is advanced to suggest that the convergence rate of BOAP will be the maximum of the convergence rates that one would find with the augmented and unaugmented GP models in isolation

**Weaknesses:**

- The concatenation of different GP models, using some as inputs to others, is not rigorously investigated. What happens when kernel lengthscales are dynamically set based on a different model at different inputs? Is this provably a positive definite kernel still?
- The approach is based on modelling both the objective functions and the user-derived feedback given $\mathbf{x}$ where neither are yet observed. What assumptions tell us that predicting the $\omega$ properties first and then predicting $y$ is easier than predicting $y$ directly? In the synthetic experiments this is clear- the functional relationship between $y$ and the $\omega$ is somehow simpler than the relationship between $y$ and $\mathbf{x}$ directly. It would be nice to clarify and understand these assumptions more carefully.
- The theory quoted in this paper from Russo & Van Roy, 2014 may not be directly applicable. The theory assumes the model is a GP, which is not guaranteed with this input-dependent lengthscale. The theory further assumes the GP model itself is fixed, whereas in the paper both GP hyperparameter optimization and updates to the $\omega$-GPs are applied at each step. (Cf. Section 6.3 of the Russo & Van Roy paper)
- The real-world experiments synthesise human feedback by converting some columns of real data into unobserved rank data. No actual human-in-the-loop experiment is conducted

**Questions:**

- what existing work has been done on using GP-predicted values as inputs to another GP? Has anyone studied this model? Do we know if it is actually a (mathematical) GP?

---

> ### Author Response · Authors · 2023-11-22
> **Response to reviewer NPJR**
>
> We thank our reviewer's time and effort invested in understanding the paper to raise some interesting points.
>
> - _**GP-predicted values as inputs to another GP? Has anyone studied this model?**_
>
>     - Using GP predictive distribution to model another GP has been explored in the past. [1] proposed a two-layered hierarchical model for GP regression that partitions the input data points such that the upper layer is responsible for coarse modelling and the lower layer is responsible for fine modelling.
>     - [1] Park, Sunho, and Seungjin Choi. \textquotedbl Hierarchical Gaussian process regression.\textquotedbl{} In Proceedings of 2nd Asian Conference on Machine Learning, pp. 95-110. JMLR Workshop and Conference Proceedings, 2010.
>
> - _**Why predicting the properties fìrst and then predicting $y$ is easier than predicting directly?**_
>     - The abstract properties can be simple physical properties or even a combination of multiple physical properties that an expert uses
> to reason about why one design is better than another. A well-chosen abstract property captures the input space of a system sufficiently
> enough that enable the domain expert to better reason about the output of the system in terms of such higher-order abstract properties. The main objective of augmenting abstract properties into the input space is to capture the inherent transformations of the function space $\mathcal{F}$ that significantly reduce the complexity and thereby simplify the
> task of GP modelling, thus a better overall optimization performance.
>     - Furthermore, human expert feedback is not available always, thus we cannot simply use expert-derived feedback as input. However we model this (intermittent) expert feedback using a GP, and that model will act as a proxy for the expert in experiments where no expert feedback is given.
>
> - _**The theory quoted in this paper from Russo \& Van Roy, 2014 may not be directly applicable?**_
>
>     - At every step of a typical BO, the kernel hyperparameters are updated, so technically speaking the model is different at each step. Nevertheless, the results of papers like Russo and Van Roy, while not precisely matched to reality, are considered \textbf{indicative} of performance, typically in big-O form (no one expects exact predictions, just an indication of expected performance). The only real distinction in this case is that our kernel - which in effect includes the posterior mean/variance of the expert models in the kernel definition - varies more radically in the early stages of optimization. We discuss Russo and Van Roy in this sense - a guide to what we might expect, not a limiting framework.
>
> - -**Dynamically setting kernel lengthscales based on a different model at different inputs? Is this provably a p.d kernel still?**_
>     - Spatially varying lengthscales have already been explored in the past literature. [2] proved the positive definiteness of the spatially
> varying kernel by showing that it has the required property of spatial variation of the lengthscales by suitably replacing the constant lengthscale with an arbitrary parameterized function of x. We refer our reviewer to \textquotedblleft Non-stationary Covariance Functions\textquotedblright{} Section (2.3.2) of [2] for the detailed discussion on spatially varying lengthscales and the resultant positive definite kernels.
>
>     - [2] Gibbs, Mark N. "Bayesian Gaussian processes for regression and classification", PhD diss., University of Cambridge, 1998.
>
> - _**No actual human-in-the-loop experiment is conducted**_
>    - Level of expertise required to conduct these experiments is high and access to such real experts is very difficult. Due to the cost and access limitations of experimental facilities, we rely on numerical simulations and simulated data.

---

> > ### Comment · Reviewer_NPJR · 2023-11-23
> > **Reply to authors' response**
> >
> > > GP-predicted values as inputs to another GP? Has anyone studied this model?
> >
> > Thank you for your answer. The work by Park et al. seems a bit different to what you do- specifically their equations (7)-(8) imply that inputs $\mathbf{c}$ are used to set the mean for a second GP using inputs $\mathbf{x}$, whereas in the submission you augment $\mathbf{x}$ to $\mathbf{\tilde{x}}$ by concatenation. This is not necessarily a problem, but I think more explanation around this would be very beneficial to the paper.
> >
> > > Why predicting the properties fìrst and then predicting $y$ is easier than predicting directly?
> >
> > Thank you for your reply, which makes sense on an intuitive level. I was wondering whether you could codify your assumptions more formally, but I take it that this is not possible.
> >
> > > Dynamically setting kernel lengthscales based on a different model at different inputs? Is this provably a p.d kernel still?_
> >
> > Thank you for this reference. I had a quick read of the paper and indeed it seems that it is always permissible to set the kernel matrix in a Gaussian kernel to $(\Sigma_i + \Sigma_j)/2$, which applies to your case with diagonal matrices. I do suggest including a reference to this

---

### Author Response · Authors · 2023-11-22
**Common Response to all Reviewers**

We thank our reviewers' efforts to raise important points and provide valuable suggestions.

- _**How do extra dimensions aid in the optimization process?**_

    - As per the definition of the eluder dimension (Definition 3.1 and Definition 3.2) provided in the main paper, it is a measure of the complexity of the function space $\mathcal{F}$ containing $f$. The eluder dimension corresponds to the number of independent and sequential evaluations needed to sufficiently approximate the given objective function $f$ within the $\epsilon$ error bound. It is reasonable to assume that as we accumulate data points, the accuracy of the human-augmented arm will increase, thereby requiring a lesser number of data points to achieve a given $\epsilon$ error bound and thus reduce the eluder dimension over time.

- _**Construction of Preference Data $P$ and their updates?**_

    - $X =\\{\mathbf x_{i} | \forall i \in \mathbb{N_{\text{n}}} \\}$, be a set of n training instances. Let $\omega=\omega(\mathbf{x})$ be the immeasurable latent preference function values associated with each of the instances $\mathbf{x}\in X$. We assume an expert compares and prefers an instance over another based on the utility $\omega$. Let $P$ be the set of $p$ pairwise preferences between instances in $X$, defined as $P^{\omega_{i}}=\\{(\mathbf{x}\succ\mathbf{x}')\quad\text{if}\;\omega_{i}(\mathbf{x})>\omega_{i}(\mathbf{x}'),\mathbf{x}\in X,\forall j\in\mathbb{N}_{P}\\}$.


    - We generate a preference list $P^{\omega_{i}}$ for each high-level feature of the designs by comparing its utility $\omega$. We initially start with $\binom{t'}{2}$ preferences in $P$. Preference relations are updated accordingly for each candidate suggested at the end of $i^{th}$ iteration. Here, we randomly choose a subset of data points containing $n/2=(t'+i)/2$ data points to compare with the new candidate suggested for function evaluation. With the updated observation model $\bar{D}=\\{\mathbf{x_{\text{{1:n}}}},P=\\{(\mathbf{x} \succ \mathbf{x}\prime)| \forall\mathbf{x},\mathbf{x}'\in X\\}$. We tune the hyperparameters ($\theta_{h}$) of the kernel and refit $\mathcal{GP}_{h}$.

- _**Maximum Predictive Likelihood**_

    - If we maximize the log marginal likelihood using all the training instances then there are chances that only the Control arm may get selected in the majority of the rounds, because of the explicit evaluations. Therefore instead of using all of the training data at once for computing the log marginal likelihood, we use only the subset of the original training data for finding the optimal hyperparameter set $\theta_{h}$ and then we use the held-out instances from the original training set to compute the (predictive) likelihood given by the equation $\log\mathcal{L}=-\frac{1}{2}(\mathbf{y}^{\intercal}(\mathbf{K}+\sigma_{\text{GN}}^{2}\mathbf{I})^{-1}\mathbf{y})-\frac{1}{2}\log|\mathbf{K}+\sigma_{\text{GN}}^{2}\mathbf{I}|-\frac{n}{2}\log(2\pi)$

    - At each iteration $t$, we compare the abovementioned predictive likelihood of both the augmented arm and the un-augmented control arm to choose the model to use for suggesting the next promising candidate.

---

### Meta-Review · Area_Chair_VYzi · 2023-12-23

**Metareview:**

This work presents a new method for Bayesian optimization that augments the surrogate model with latent attributes estimates via preference learning. The authors propose a kernel for this augmented model, and a heuristic for improving the robustness of the optimization algorithm to misspecification (or lack of utility) due to the inclusion of the latent attributes.  Reviewers found the setup to be interesting.  However, they were not satisfied with the justification of the kernel, design choices in the algorithm, and the theoretical results. Some reviewers felt that the experiments should more rigorously evaluate choices about how often the DM is queried (hhJX, 4RNk). Reviewer 4RNk also pointed out that various preference elicitation strategies could be used and the effect of varying levels of noise or different noise models is not examined. From the references, I presume the reviewer was referring to considering noise wrt the latent utility, rather than the coin flip used in the submission. During the discussion phase, not all questions or critiques were addressed.  The paper received no accept ratings, and will thus be rejected.  I recommend trying to address these pieces of feedback in any future submissions of this work.

Augmenting the input space with latent features learned from DMs is a neat idea, and I enjoyed reading the paper. Here are a few of my own thoughts on the work:
- Why not model uncertainty in the augmented latent features as input noise?  This could be accomplished easily via a Monte Carlo acquisition function. It is possible that there is some relationship between such a scheme and the proposed kernel.
- Why select a single model based on the likelihood rather than perform Bayesian model averaging (i.e., do a weighted average of the models according to their likelihoods)?
- Why use TS? EI tends to work better in practice and appears to do so in your benchmarks, when no auxiliary information is available. The existing setup (and integrating over input uncertainty) would be easy to implement via a MC EI with sample average approximation (e.g., in BoTorch).
- The authors report one standard error instead of conventional two SEs (the 95% interval). Many of the results would are not statistically significant and this would be more apparent if the authors reported 2 SEs. I would recommend running at least 20 replications, as is standard in the field.
- The experiments appear limited to fairly low-dimensional problems. It would be nice to see this in higher dimensions. Perhaps some of the test functions used in the PBO literature or works involving compositional objectives could be utilized.
- Unless the authors can propose a model that reduces to a stationary GP, I'd recommend removing the theory / references to the Russo & Van Roy paper, as they are not applicable to this setting.

**Justification For Why Not Higher Score:**

Publication is not yet ready. A lot of open questions about design choices, more sensitivity analysis needed, etc.

**Justification For Why Not Lower Score:**

N/A

---

### Decision · Program_Chairs · 2024-01-16

Reject